# Dual-stream Perception-driven Blind Quality Assessment for Stereoscopic Omnidirectional Images

## ABSTRACT

The emergence of virtual reality technology has made stereoscopic omnidirectional images (SOI) easily accessible and prompting the need to evaluate their perceptual quality. At present, most stereoscopic omnidirectional image quality assessment (SOIQA) methods rely on one of the projection formats, i.e., Equirectangular Projection (ERP) or CubeMap Projection (CMP). However, while ERP provides global information and the less distorted CMP complements it by providing local structural guidance, research on leveraging both ERP and CMP in SOIQA remains limited, hindering a comprehensive understanding of both global and local visual cues. Motivated by this gap, our study introduces a novel dual-stream perception-driven network for blind quality assessment of stereoscopic omnidirectional images. By integrating both ERP and CMP, our method effectively captures both global and local information, marking the first attempt to bridge this gap in SOIQA, particularly through deep learning methodologies. We employ an inter-intra feature fusion module, which considers both the inter-complementarity between ERP and CMP and the intra-relationships within CMP images. This module dynamically and complementarily adjusts the contributions of features from both projections and effectively integrates them to achieve a more comprehensive perception. Besides, deformable convolution is employed to extract the local region of interest, simulating the orientation selectivity of the primary visual cortex. Finally, with the features of left and right views of SOI, a stereo cross attention module that simulates the binocular fusion mechanism is proposed to predict the quality score. Extensive experiments are conducted to evaluate our model and the state-of-the-art competitors, demonstrating that our model has achieved the best performance on the databases of LIVE 3D VR, SOLID, and NBU.

## CCS CONCEPTS

• **Computing methodologies** → *Simulation evaluation*;

## KEYWORDS

Stereoscopic Omnidirectional Images, Image Quality Assessment, Virtual Reality, Stereoscopic Visual Perception, Visual Experience Quality Assessment

*ACM MM, 2024, Melbourne, Australia*
© 2024 Copyright held by the owner/author(s). Publication rights licensed to ACM.
ACM ISBN 978-x-xxxx-xxxx-x/YY/MM
https://doi.org/10.1145/nnnnnnn.nnnnnnn

## 1 INTRODUCTION

With the rapid development of Virtual Reality (VR) technology and the popularization of head-mounted displays, people have been able to apply VR technology in various fields, such as leisure and entertainment, business services, and healthcare [42]. The growing demand for immersive, 3D viewing is driving developers to provide better experiences [9]. However, distortion inevitably occurs during image acquisition, compression, transmission, and reconstruction, affecting the user's visual experience [39]. To ensure better subjective perception quality, it is essential to quantify the extent of image quality degradation, enabling image optimization and quality control. Stereoscopic omnidirectional image quality assessment (SOIQA) methods closely resemble those applied to 2D images and could be categorized into subjective and objective quality assessment [22]. Although subjective assessment is close to human perception, it is time-consuming, costly, and difficult for standardization. Therefore, there is an urgent need to study objective assessment methods that are consistent with subjective evaluation to accurately predict the perceived quality of the stereoscopic omnidirectional image (SOI).

In VR systems, before encoding, it is necessary to project omnidirectional images (OI) onto a 2D plane, and equirectangular projection (ERP) is often considered as the default projection format. The projection method maps the longitude lines of the sphere into equally spaced vertical lines, and maps the sphere's latitudinal lines into equally spaced horizontal lines, forming a rectangular plane that can effectively retain the global information of the OI. However, there is significant geometric distortion in the polar regions of the ERP image caused by the stretching [20]. Meanwhile, cubemap projection (CMP) is a projection method that involves projecting spherical content onto a cubic model, unfolding each face, and then stitching them together into a rectangular format. Unlike ERP, CMP projects spherical content onto six faces, which effectively solves the problem of projection distortion of the north and south poles, but still has the over-sampling problem within the edge of each surface. In a nutshell, ERP provides global information and the less distorted CMP complements it by providing local structural guidance [7][11]. The combination of ERP and CMP allows for a more comprehensive utilization of information in OI, encompassing both global and local aspects. In this way, the integration enables the complementary advantages of the two projection formats, collectively providing a complete visual perception of OI.

Given that ERP offers comprehensive global information while CMP is considered more localized, the extraction of complementary features from both formats is of utmost importance. Convolutional Neural Networks (CNN) can effectively extract features from local window images based on their shared convolutional kernel weights and local inductive bias, which has been applied widely in various computer vision tasks. Nonetheless, due to the limited receptive field, CNN still fails to explicitly model the long-range relationship.

Meanwhile, Transformer [27] relies on its self-attention mechanism to be able to assign weights to all image blocks to fully extract the global features of the entire image. While the Transformer effectively captures long-range dependencies in global features, it tends to overlook local feature information. Considering that human visual perception is a hierarchical structure, processing visual signals from local details to global information, neither local nor global features alone can comprehensively represent the true visual perception [21]. The combination of local and global perceptual quality measurement aligns more closely with the human visual system (HVS). Hence, employing both CNN and Transformer as a dual-branch backbone is essential to extract both local and global feature information.

Considering that both ERP and CMP features exhibit information complementarity and content redundancy, effectively achieving their complementary fusion is another crucial issue to be addressed. Indiscriminate fusion strategies, such as concatenation convolution or addition, would fail to fully leverage the complementary relationship between the two, thus hindering the formation of a complete visual perception. The inter-complementarity between ERP and CMP, and the intra-relationships within CMP images need to be thoroughly explored in the fusion process. Furthermore, the issue of left-right view fusion also needs to be considered for SOIQA. Binocular stereoscopic vision involves a complex visual interaction, encompassing both binocular fusion and rivalry[30]. A key challenge in stereoscopic vision lies in the correspondence process. To analyze and integrate features from both left and right views, the cross-attention mechanisms are usually employed [27]. However, it's crucial to tailor these mechanisms to account for the distinct characteristics of left-right views.

To address the above challenge of SOIQA, we present a novel Dual-Stream Perception-Driven Network (DSPDNet), which could optimally leverage the complementary advantages of ERP and CMP. DSPDNet consists of three modules: dual-stream feature extraction module (DFM), inter-intra feature fusion module (FFM), and stereo cross attention module (CAM). Specifically, in the DFM, the feature extractor for local information in CMP is a modified ResNet in which deformable blocks [8] are explored to mimic the orientation selectivity of HVS, and could adaptively select the region of interest. Simultaneously, the feature extractor for global information in ERP is based on the Swin Transformer. The role of inter-intra FFM is to combine the global features extracted by ERP with the local features extracted by CMP, allowing both feature sets to fully exploit their mutually beneficial complementary functions. To this end, the squeeze-and-excitation (SE) block for channel attention is applied on ERP and CMP to obtain six CMP-induced inter features, and then integrate them with the ERP feature via the adaptive weights calculated by the intra-relationships within CMP images. Finally, motivated by the cross-attention mechanism, the CAM fully leverages and integrates feature information from both the left and right views, allocating weights to better simulate the binocular fusion and rivalry mechanism. Unlike the traditional cross-attention mechanism that scans all image locations, the CAM specifically focuses on corresponding features along the horizontal disparity line, ensuring high flexibility while capturing global correspondence.

In summary, the principal contributions of our paper are summarized as follows:

- To the best of our knowledge, the proposed DSPDNet is the first study that leverages the complementary advantages of ERP and CMP via deep learning ways for SOIQA. Specifically, a hybrid CNN-Transformer feature extraction network is built to learn the perceptual representation considering local details and global information.
- For a more comprehensive utilization of local and global features of ERP and CMP, an inter-intra feature fusion module is employed accounting for the inter-complementarity between ERP and CMP and the intra-relationships within CMP images. Additionally, deformable convolution is employed to adaptively select the region of interest.
- Motivated by the cross-attention mechanism, a stereo cross attention module (CAM) helps the two view images of SOI learn to obtain mutual attention maps representing the correlation between them, which could simulate the process of human binocular correspondence process.

## 2 RELATED WORK

The stereoscopic omnidirectional image (SOI) has the characteristics of both the stereoscopic image (SI) and the omnidirectional image (OI). Therefore, a brief overview of the SI, OI, and SOI quality assessment methods is given in this section.

### 2.1 Stereoscopic Image Quality Assessment

Earlier stereoscopic image quality assessment (SIQA) methods usually take a two-stream architecture that processes the left and right view images separately. Yasakethu *et al.* [36] apply 2D metrics including PSNR, and SSIM to the two views, and average them for the quality score at the end. To further increase the accuracy, Sim *et al.* [24] utilize deep CNN to extract the semantic features and quality-aware features of both views, and feed them straight into support vector regression (SVR) to get the quality score.

Inspired by the theory of binocular perception, recent interests focus on how to utilize the correlation between the left and right view of stereoscopic images in SIQA. Zhang *et al.* [38] propose to add the difference maps between the two views as the inputs to the CNN, aiming at taking into account the variability between binoculars. Besides difference maps, Zhou *et al.* [40] find the performance can be further improved by adding the fusing maps of the two views. As the information interactions within the human visual system (HVS) cannot be adequately captured by the simple difference/fusion maps, a new approach involving binocular interaction modules [23] is introduced. This module incorporates a new form of cross-convolution to model binocular interactions within visual cortical regions, enabling the extraction of more semantic inter and intra features from the two views. However, these methods only perform well on 2D images, which cannot be directly applied to omnidirectional images.

### 2.2 Omnidirectional Image Quality Assessment

Different from stereoscopic images (SI), omnidirectional image (OI) provides the observer with a spherical viewing range based on all positions with an infinite field of view. In practice, projecting the original spherical image into a 2D plane, i.e., equirectangular projection (ERP) for encoding is a common choice. Therefore, most

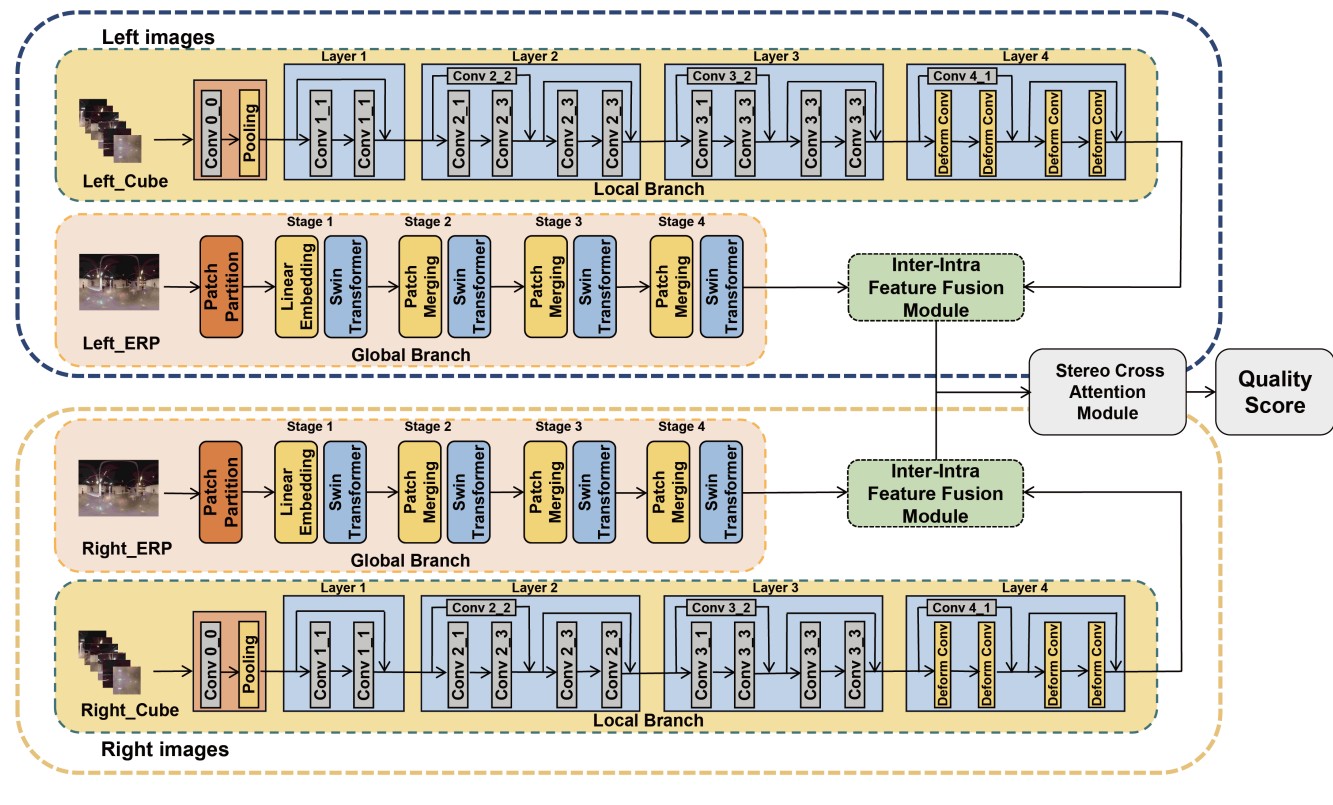

Figure 1: The framework of the proposed method.

early omnidirectional image quality assessment (OIQA) studies are conducted based on ERP. Liu *et al.* [16] extract luminance features, global entropy, and color features on ERP, and finally get the quality score by support vector regression (SVR) based on the above multiple features and human subjective quality evaluation. Meanwhile, Kim *et al.* [14] partition the ERP into patches and evaluate it by the patch's positional features, estimating the weights and quality scores of the patches, and then aggregating the weights and scores of all patches to predict the image quality scores. Noticing that patches sampled from the ERP image contain heavy geometry deformation, viewports-based OIQA methods [26, 32] have recently attracted much attention.

While effectively preserving the global information of the omnidirectional image, the conversion to ERP format by the sphere-to-plane projection introduces a lot of geometric distortion in the polar regions. To this end, cubemap projection (CMP) that projects the sphere on six surfaces of the cube is studied and first introduced into OIQA by Jiang *et al.* [13]. Later, they design a color omnidirectional distortion unit [12] composed of multiple color CMP images, trying to simulate the user's viewing behavior in OIQA. Compared to ERP, the distortion of CMP is not so serious, but the six planes may divide the object in an omnidirectional image into several parts, resulting in object discontinuity [7]. One recent interest in OIQA is how to utilize the complementarity of these two projection modes ERP and CMP. Qiu and Shao [20] present an OIQA model that is pre-trained with local information in CMP and later fine-tuned with

global information in ERP. However, the content redundancy and information complementarity of ERP and CMP are not carefully investigated, which is very important when taking them as two source inputs for OIQA tasks.

## 2.3 Stereoscopic Omnidirectional Image Quality Assessment

Stereoscopic omnidirectional image (SOI) usually consists of the left and right view images, both of which are omnidirectional images (OI). Quality assessment for SOI has been a relatively recent emerging topic. Yang *et al.* [35] study the latitude characteristics of ERP images and propose an improved OI saliency model, by which more meaningful features of left and right view images are extracted and then fused by tensor decomposition for predicting the quality score. Differently, Poreddy and Appina [18] convert an ERP image into six CMP images to overcome the aforementioned distortion. Using CMP features, a bivariate generalized Gaussian distribution model is presented to model the joint dependence between luminance and disparity of two views. Zhou *et al.* [41] design a projection invariant feature learned from 6 kinds of projection formats (ERP, CMP, etc.), and prove its effectiveness in SOIQA tasks. However, the above methods are based on hand-designed features. Although visual saliency is often adopted to facilitate prediction accuracy, the hand-designed features still have limited generalization and flexibility.

To further improve prediction accuracy, researchers have begun to introduce deep neural networks into SOIQA recently. Yang *et al.* [34] first imply the depth information of ERP images, and then utilize spherical CNN [6] for SOIQA to consider the spherical viewing characteristics. Noticing that it is still unclear whether spherical CNN can be well adapted to feature extraction of ERP images, Chai *et al.* [1] propose a monocular and binocular interaction-oriented three-channel network architecture for SOIQA, in which the difference map of two views is utilized for fusion and deformable convolution is adopted to ensure the invariant receptive fields of convolutional kernels on ERP images.

From the recent achievements of SOIQA, research on utilizing multi-source projection formats of omnidirectional images is insufficient, especially via deep learning ways. To fill the gaps, our study investigates the complementary of ERP and CMP images, and proposes a dual-branch backbone with Transformer and deformable CNN to simultaneously extract the global features from ERP and local features from CMP, which are later integrated and refined for predicting the quality score. Ablation studies show the integration of local and global perceptual features has achieved obvious performance improvement.

## 3 THE PROPOSED METHOD

In this section, we first introduce the general architecture of the method, and then discuss the details of its main modules.

The overall architecture of our model is shown in Fig. 1. For a left/right view of a stereoscopic omnidirectional image (SOI), its ERP image and six CMP images are used together as input. The ERP image provides global features extracted by Swin Transformer, while the CMP images provide structural guidance as a supplement. We utilize ResNet with deformable convolution blocks instead of conventional convolution to extract these local features from CMP images. The global and local features are then adaptively fused by an inter-intra feature fusion module, which enables automatic learning of the correlation and significance between features derived from both projection modes. Finally, a stereo cross-attention mechanism is employed to simulate the interaction between the left and right view images, mapping the features to obtain an objective quality score.

### 3.1 Image Pre-processing

Usually, in a stereoscopic omnidirectional database $O$, the raw data is stored in both left ERP images $I_L$ and right ERP images $I_R$. The following modules will be introduced using $I_L$ as an example, and the same operations will be applied to $I_R$.

For an ERP image $I \in R^{C*H*W}$ in $I_L$, severe distortion has inevitably existed in the bipolar region as shown in Fig.2-(a), which deforms the object and affects the assessment of its quality. We transform $I$ and obtain its six CMP images $\{CP_i \in R^{C*H*W}\}_{i=1}^6$ as shown in Fig.2-(b). It can be seen that CMP images have reduced the distortion in ERP. For the convenience of training, we resize the ERP image $I$ and CMP images $\{CP_i\}_{i=1}^6$. In experiments, $H$ and $W$ are 256, and $C$ is 3.

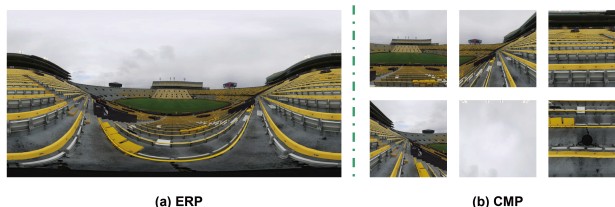

**(a) ERP**  **(b) CMP**

**Figure 2: Comparison of ERP and CMP image distortion in the same scene.**

### 3.2 Dual-stream Feature Extraction Module

*3.2.1 The Local Branch.* The input images of the local branch are the CMP images $\{CP_i\}_{i=1}^6$. ResNet34 [10] is chosen as the backbone network for feature extraction, which can obtain very rich local spatial information with fewer model parameters. Specifically, the first layer contains a 7x7 convolutional layer and a max pooling. The following are 4 residual layers that consist of (3,4,6,3) Drop-Blocks in each layer, respectively. Each DropBlock contains two 3x3 convolution layers with slightly different ways of connecting residuals, each followed by a batch normalization layer and a ReLU activation function.

In ResNet34, a deformable convolution block is introduced to model the response of the human visual system (HVS) to regions of interest, as illustrated in Fig. 1. Specifically, in the last residual layer of ResNet34, all 3x3 convolutions within the three dropblocks are replaced with deformable convolutions. The deformable convolution consists of a ordinary convolutional layer OConv and a deformable convolutional layer DConv. OConv is used to learn an offset map $\Delta p$ from the input features. With the learned $\Delta p$, the subsequent DConv Block realizes adapting to different object shapes and geometries through Eq.(1), thereby emphasizing the human perception of objects.

$$DF_{C_i} = DConv(F_{C_i}, \Delta p) \tag{1}$$

Upon obtaining the deformable CNN features $\{DF_{C_i} \in R^{C*P*P}\}_{i=1}^6$, we take them as the local feature $F_{local}$ of the six CMP images, since they are the different views of the stereoscopic omnidirectional images. In experiments, $C$ and $P$ are 512 and 8, respectively.

*3.2.2 The Global Branch.* The input image of the global branch is the ERP image $I$. Swin Transformer [17] is chosen as the backbone network for global feature extraction. Specifically, the image $I$ is firstly divided into non-overlapping patches. Then, linear embedding is performed to change the number of channels in the patch to fit the model's requirements. The patches are processed by the Swin Transformer modules to extract descriptive feature maps by window self-attention and shifted window self-attention, enabling information exchange between different windows. A patch merging module is used after a Swin Transformer layer to reduce the spatial size of the feature map while increasing the number of channels. In this way, the window gradually increases the receptive field, allowing the extraction of global contextual features. The output of the global branch is $F_{global} \in R^{C*P*P}$.

Figure 3: Architecture of inter-intra feature fusion module (FFM).

## 3.3 Inter-intra Feature Fusion Module

As mentioned above, ERP feature $F_{global} \in R^{C*P*P}$ provides global information and the less distorted CMP features $F_{local} = \{DF_{C_i} \in R^{C*P*P}\}_{i=1}^{6}$ complement it by providing local structural guidance. The combination of $F_{global}$ and $F_{local}$ allows for a more comprehensive utilization of information in SOI, encompassing both global and local aspects. Due to the interconnection among various unfolded faces of the CMP and their differing importance for quality assessment tasks, it is necessary to consider both the intra fusion within the CMP and the inter fusion between the ERP and CMP. To this end, we utilize an inter-intra feature fusion module (FFM) to dynamically and complementarily integrate the local CMP feature $DF_{C_i} \in R^{C*P*P}$ with the global ERP feature $F_{global} \in R^{C*P*P}$. The structure of FFM is illustrated in Fig. 3.

Firstly, the interfusion of CMP and ERP features are considered to automatically pick out the valuable parts of each projection. We concatenate $DF_{C_i}$ and $F_{global}$ along the channel dimension and apply it to a squeeze-and-excitation (SE) block for channel attention which is composed of two linear layers, one ReLu layer, and one sigmoid activation function. A bottleneck convolutional layer and a sigmoid activation function are followed after the SE block to evaluate how much local information should be provided for the global features. The interfusion computation process is formulated in Eq. (2).

$$P_i = \sigma(conv(SE(DF_{C_i}, F_{global}))) \quad (2)$$

where $P_i$ represents the contribution of $DF_{C_i}$. Thus, the fused feature that could be obtained as follows.

$$F_i = P_i \cdot DF_{C_i} + (1 - P_i) \cdot F_{global} \quad (3)$$

By Eq. (3), the six CMP images from different perspectives offer six induced fusion features, each contributing differently and holding varying importance in the final fusion. To emphasize pertinent information and mitigate irrelevant redundancies, further learning of adaptive weights for intra features is necessary. Firstly, the original CMP features $\{DF_{C_i} \in R^{C*H*W}\}_{i=1}^{6}$ are concatenated, and a SE block is employed to obtain a vector $\alpha \in R^{6C*1*1}$. This vector

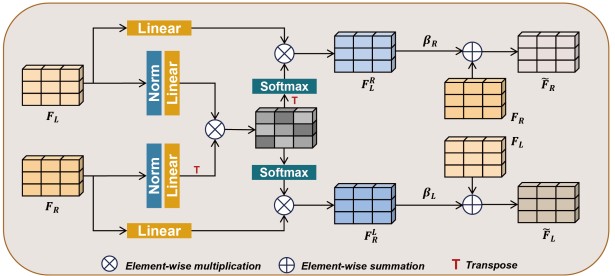

Figure 4: Architecture of stereo cross attention module (CAM).

$\alpha$ is then divided into six vectors $\{\alpha_i \in R^{C*1*1}\}_{i=1}^{6}$. By normalizing these vectors, we can obtain dynamic weights $\omega_i$. The above process can be represented by Eq. (4), (5).

$$\alpha_i = Split(SE(Con(DF_{C_1}, DF_{C_2}, ..., DF_{C_6}))) \quad (4)$$

$$\omega_i = \frac{Sum(\alpha_i)}{\sum_{i=1}^{6} Sum(\alpha_i)} \quad (5)$$

where $Con(\cdot)$ represents channel concatenation, $Split(\cdot)$ represents the operation of splitting a vector into six vectors, and $Sum(\cdot)$ represents the summation of all elements in a vector. Ultimately, we fuse the six CMP-induced inter features $\{F_i\}_{i=1}^{6}$ via the adaptive weights $\{\omega_i\}_{i=1}^{6}$, and integrate them with the ERP feature $F_{global}$ to obtain the final fused feature $F_L$ for the left view, as specified in Eq. (6).

$$F_L = F_{global} + \sum_{i=1}^{6} \omega_i \cdot F_i \quad (6)$$

## 3.4 Stereo Cross Attention Module

Upon obtaining the features $F_L$ and $F_R$ of the left and right views of SOI respectively, we discuss how to use the two features to evaluate the quality of the SOI in this section. Unlike just the weight-sum of the representation of different views, we propose a stereo cross attention module (CAM) shown in Fig. 4 to calculate the relevance of cross-view features, aiming to simulate the binocular fusion mechanism of the human visual system (HVS).

Specifically, the features $F_L$ and $F_R$ are firstly normalized as $\overline{F}_L = LN(F_L)$ and $\overline{F}_R = LN(F_R)$ via layer normalization (LN). Then, the bidirectional cross-attention is calculated using scaled dot-product attention (SDPA) [27] between left-right views by Eq. (7), (8).

$$F_R^L = Attention(W_1^L \overline{F}_L, W_1^R \overline{F}_R, W_2^R F_R) \quad (7)$$

$$F_L^R = Attention(W_1^R \overline{F}_R, W_1^L \overline{F}_L, W_2^L F_L) \quad (8)$$

Here, $W_1^L$, $W_1^R$, $W_2^L$, and $W_2^R$ are projection matrices. For $F_R^L$, $W_1^L \overline{F}_L$ is the query matrix projected by left-view feature, and $W_1^R \overline{F}_R, W_2^R F_R$ are the key and value matrices projected by right-view feature. Unlike the traditional cross-attention mechanism, we use the same query matrix and key matrix in $F_R^L$ and $F_L^R$ to represent

Table 1: Performance of competitors and our method on three databases.

| Types | Approaches | LIVE 3D VR Database | | | SOLID Database | | | NBU Database | | |
|-------|-----------|-------|------|------|-------|------|------|-------|------|------|
| | | SROCC | PLCC | RMSE | SROCC | PLCC | RMSE | SROCC | PLCC | RMSE |
| 2D-IQA | PSNR | 0.6551 | 0.5948 | 9.1950 | 0.8471 | 0.8629 | 0.3996 | 0.7809 | 0.7828 | 0.5934 |
| | SSIM[28] | 0.6127 | 0.6744 | 8.4458 | 0.8664 | 0.8507 | 0.4089 | 0.7997 | 0.7902 | 0.5843 |
| | MS-SSIM[29] | 0.5477 | 0.5582 | 9.5082 | 0.7550 | 0.7730 | 0.6430 | 0.8604 | 0.8544 | 0.4954 |
| SIQA | Chen[3] | 0.7509 | 0.7815 | 7.0807 | 0.8771 | 0.8472 | 0.4100 | 0.8495 | 0.8749 | 0.4408 |
| | SINQ[15] | 0.8191 | 0.8207 | 6.4536 | 0.7794 | 0.8105 | 0.4602 | 0.7620 | 0.8030 | 0.4960 |
| | PAD-net[33] | 0.6983 | 0.7057 | 5.6581 | 0.7672 | 0.7887 | 0.8663 | 0.8426 | 0.8270 | 0.8260 |
| OIQA | S-PSNR[37] | 0.6344 | 0.6865 | 8.1639 | 0.8564 | 0.8700 | 0.3892 | 0.8250 | 0.8080 | 0.5620 |
| | WS-PSNR[25] | 0.6125 | 0.6103 | 8.9940 | 0.8512 | 0.8177 | 0.4352 | 0.8020 | 0.7910 | 0.5840 |
| | VGCN[32] | 0.8429 | 0.8368 | 5.4584 | 0.6331 | 0.6213 | 0.6970 | 0.6416 | 0.7207 | 0.6398 |
| SOIQA | VP-BSOIQA[19] | 0.8013 | 0.7956 | 6.8481 | 0.8420 | 0.8537 | 0.4115 | 0.8600 | 0.8830 | 0.4100 |
| | Chai-SOIQE[1] | 0.8812 | 0.8637 | 6.2359 | 0.8722 | 0.8799 | 0.3720 | 0.9336 | 0.9465 | 0.2992 |
| | SOIQE[4] | 0.6575 | 0.6795 | 8.3917 | 0.9240 | 0.9270 | 0.3830 | 0.9095 | 0.9197 | 0.3583 |
| | Proposed | 0.9378 | 0.9342 | 4.0805 | 0.9613 | 0.9681 | 0.2331 | 0.9776 | 0.9846 | 0.1515 |

each intra-view feature and calculate their correlation along the width dimension because the left and right views of SOI are highly symmetric under epipolar constraint [5]. SDPA is an efficient attention mechanism to calculate the attention scores between the query and the key-value pair in Eq. (9).

$$Attention(Q, K, V) = Softmax(\frac{QK^T}{\sqrt{C}})V \qquad (9)$$

Finally, the cross attention-based features $F_R^L$ and $F_L^R$ are fused with the features $F_L$, $F_R$ through Eq. (10) and Eq. (11), respectively, to obtain the final features of left and right views.

$$\tilde{F}_L = \beta_L \cdot F_R^L + F_L \qquad (10)$$

$$\tilde{F}_R = \beta_R \cdot F_L^R + F_R \qquad (11)$$

where $\beta_L$ and $\beta_R$ are trainable channel-wise weights which represent the contribution of the cross attention-based features. Based on the cross-view features $\tilde{F}_L$ and $\tilde{F}_R$, the quality score $Q$ is predicted by a fully connected layer using Eq. (12).

$$Q = \frac{FC1(\tilde{F}_L) + FC2(\tilde{F}_R)}{2} \qquad (12)$$

## 4 EXPERIMENTAL RESULTS AND ANALYSIS

In this section, we discuss the experimental results of our method (DSPDNet). Firstly, we describe the databases and evaluation criteria. Next, we compare the performance of our method with the state-of-the-art ones. Finally, ablation experiments are performed to investigate each component of the DSPDNet.

### 4.1 Experimental Settings

1) **Databases:** We conduct experiments on three publicly available SOI databases, namely LIVE 3DVR [2], SOLID [31], and NBU [19].

**LIVE 3D VR** database consists of 15 original images and 450 distorted images, including six types of distortion such as Gaussian blur, Gaussian noise, downsampling, stitching artifacts, VP9 compression, and H.265 compression. In our experiments, we select 449 distorted images for the training and testing processes. Following the suggestion of Chen *et al.* [2], one distorted image is rejected as an outlier. Each distorted image is provided with a differential mean opinion score (DMOS) ranging from 0 to 100 as the subjective quality score. Lower DMOS values indicate better visual quality.

**SOLID** database comprises 276 distorted 3D VR images, obtained by compressing six original images with BPG and JPEG compression, resulting in three depth levels for each set of distorted images. Each distorted image is provided with a mean opinion score (MOS) ranging from 0 to 5 as the subjective score. Higher MOS values indicate better visual quality.

**NBU** database consists of 12 original images and 396 distorted images, encompassing three common distortion types, namely JPEG, JPEG2000, and HEVC. Each distorted image is provided with a MOS value ranging from 0 to 5 as the subjective score. Higher MOS values indicate better visual quality.

2) **Evaluation Criteria:** Three metrics, as the common practice, are used to make the performance analysis, including Pearson's Linear Correlation Coefficient (PLCC), Spearman's Rank Order Correlation Coefficient (SROCC), and the Root Mean Square Error (RMSE). PLCC measures the strength of the linear correlation between the predicted objective scores and the subjective scores. SROCC measures the monotonicity of the model's predicted results and serves as an effective non-linear correlation metric. The RMSE is used to assess the consistency of the model's prediction by measuring the deviation between the predicted scores and the subjective scores. Higher PLCC and SROCC values closer to 1, as well as lower RMSE values closer to 0, indicate more accurate prediction by the model.

**Table 2: Performance (SROCC/PLCC) of all models for different distortion types on LIVE 3D VR database.**

| Types | Approaches | LIVE 3D VR Database (SROCC/PLCC) | | | | | |
|---|---|---|---|---|---|---|---|
| | | Gaussian Blur | Gaussian Noise | Down Sampling | Stitching Distortion | VP9 Compression | H.265 Compression |
| 2D-IQA | PSNR | 0.7728/0.9095 | 0.8957/0.9238 | 0.8292/0.9110 | 0.6953/0.7141 | 0.5786/0.6672 | 0.7453/0.7585 |
| | SSIM[28] | 0.8114/0.9126 | 0.8935/0.9173 | 0.8114/0.8956 | 0.6422/0.7561 | 0.8135/0.8697 | 0.8536/0.8558 |
| | MS-SSIM[29] | 0.8692/0.7693 | 0.8943/0.8866 | 0.7742/0.8732 | 0.6830/0.6608 | 0.7431/0.7609 | 0.9350/0.8863 |
| SIQA | Chen[3] | 0.8571/0.8801 | 0.8957/0.9349 | 0.8464/0.8654 | 0.7179/0.7349 | 0.8628/0.8997 | 0.9436/0.9449 |
| | SINQ[15] | 0.8886/0.8810 | 0.9250/0.9337 | 0.9507/0.9432 | 0.0706/0.2939 | 0.7007/0.7168 | 0.8972/0.8682 |
| OIQA | S-PSNR[37] | 0.7757/0.8621 | 0.8786/0.9123 | 0.8500/0.8580 | 0.7223/0.7008 | 0.6750/0.7080 | 0.8393/0.8671 |
| | WS-PSNR[25] | 0.7721/0.8680 | 0.8743/0.9123 | 0.8493/0.8579 | 0.7148/0.7166 | 0.6486/0.7023 | 0.8279/0.8233 |
| SOIQA | VP-BSOIQA[19] | 0.8836/0.9201 | 0.8371/0.8387 | 0.9379/0.9237 | 0.2983/0.4407 | 0.6162/0.6131 | 0.8036/0.8111 |
| | Chai-SOIQE[1] | 0.9143/0.9401 | 0.8417/0.8530 | 0.9571/0.9584 | 0.6332/0.6255 | 0.7550/0.7543 | 0.8600/0.8526 |
| | SOIQE[4] | 0.9013/0.9406 | 0.8932/0.9093 | 0.7898/0.9126 | 0.7418/0.7784 | 0.8400/0.8552 | 0.8974/0.8950 |
| | Proposed | 0.9643/0.9881 | 0.9500/0.9779 | 0.9464/0.9825 | 0.8813/0.9124 | 0.8643/0.8602 | 0.9536/0.9768 |

**3) Implementation Details:** To save the training time, the CMP images are obtained from ERP images and stored locally beforehand. The overall model is implemented using the PyTorch framework, with ResNet34 loading the pre-trained parameters trained on ImageNet, and the Swin Transformer using randomly initialized parameters. The Adam optimizer is used during model training, with a training period of 200 epochs, a learning rate of 0.0001, and a decay rate of 0.5. The batch size is set to 8. The database is evenly divided into K groups, with each group consisting of distorted images derived from multiple original images. There are no identical image data between different groups. One group is selected as the test set, while the remaining K-1 groups are used as the training set. For the LIVE 3D VR, SOLID, and NBU databases, the values of K are set to 5, 6, and 12.

### 4.2 Comparison with the State-of-the-Arts

We test the performance of our model on three databases while comparing it with 12 state-of-the-art methods. These methods include 2D IQA methods, i.e., PSNR, SSIM [28], and MS-SSIM [29], stereoscopic IQA methods, i.e., Chen [3], SINQ [15], and PAD-net [33], omnidirectional IQA methods, i.e., S-PSNR [37] and WS-PSNR [25], and VGCN [32], and stereoscopic omnidirectional IQA methods, i.e., VP-BSOIQA [19], SOIQE [4], and Chai-SOIQE [1]. The final results are shown in Table 1, with the best performance highlighted in red. It is worth noting that all the results are obtained using the codes provided by the authors or from their published papers.

From Table 1, it can be seen that our method has achieved the best performance in the databases of LIVE 3D VR, SOLID, and NBU, i.e., with up to 8.2%, 4.4%, and 4.0% relative performance gain of PLCC, respectively. More conclusions can be drawn as follows.

The majority of methods exhibit poorer performance on the LIVE 3D VR database compared to the SOLID and NBU databases. The reason might be that the LIVE 3D VR database contains more diverse image scenes and complex types of distortion, which makes the task more challenging on this database. Owing to the complementary

properties of ERP and CMP, the performance degradation of our method is much smaller.

Besides, the scatter plot of DMOS and predicted quality scores for the five representative methods (SSIM, PSNR, MS-SSIM, SOIQE, and our method) on the LIVE 3D VR database is shown in Fig. 5. The results indicate superior convergence and monotonicity in our method compared to other competitors.

### 4.3 Comparison on Individual Distortion Type

To investigate the influence of different distortion types on the SOIQA task, we choose the LIVE 3D VR database and continue to conduct experiments on individual distortion types. In experiments, each single distortion type is selected from the test subset as the new test set, keeping other settings the same as the default. The quality scores are obtained in the end for the comparison of performance. The results are shown in Table 2, with the best performance highlighted in red and the second-best performance highlighted in blue. For the limited space, only the values of SROCC/PLCC are given in the table.

From Table 2, it can be seen that our method has always performed well, especially when dealing with the distortion caused by Gaussian blur, Gaussian noise, stitching, and H.265 compression. Take the stitching for example, SROCC increases by 18.8% from 0.7418 to 0.8813, and PLCC increases by 17.2% from 0.7784 to 0.9124. More conclusions can be drawn as follows.

No single method achieves the best performance across all distortion types. One reason might be that different distortion types have distinct characteristics. Neither deep learning features nor handcrafted feature design can fully simulate the complex perceptual process of different distortion types in HVS.

Moreover, for the stitching distortion type, the unique distortion in omnidirectional images, our method performs best and achieves the highest performance gain among all distortion types. It indicates the effectiveness of the comprehensive utilization of ERP and CMP in SOIQA, encompassing both global and local aspects.

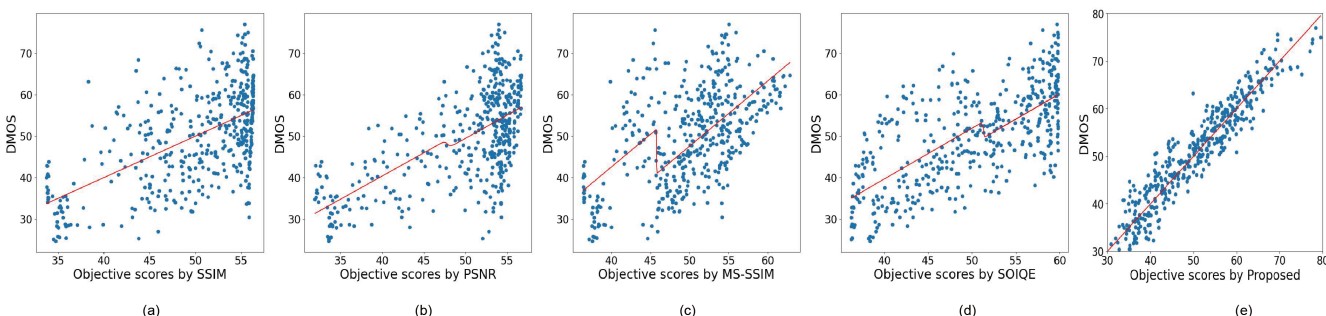

**Figure 5: Scatter plots of predicted quality scores by the five methods against the DMOS values on the LIVE 3D VR database. (a) SSIM (b) PSNR (c) MS-SSIM (d) SOIQE (e) Our method.**

Finally, by calculating the times of achieving the best performance under different evaluation metrics, it is found that our method has the highest number of hits, with a total of 10. This indicates that our method exhibits greater stability on the LIVE 3D VR database with different distortion types.

**Table 3: Ablation studies on LIVE 3D VR database.**

| Methods | LIVE 3D VR Database | | |
|---|---|---|---|
| | SROCC | PLCC | RMSE |
| *w/o* DConv. | 0.8818 | 0.8933 | 5.1402 |
| *w/o* CAM | 0.9253 | 0.9225 | 4.4129 |
| *w/o* Global | 0.9296 | 0.9266 | 4.3011 |
| *w/o* Local | 0.2876 | 0.3433 | 10.7614 |
| Full model | 0.9378 | 0.9342 | 4.0805 |

## 4.4 Ablation Study

To test the effectiveness of each component of our model, we conduct a comprehensive ablation study by disabling the corresponding components respectively. The experimental setups are the same as the default. The results are shown in Table 3.

1) **Deformable convolution:** In the proposed model, deformable convolution is used to model the response of the human visual system (HVS) to the region of interest (ROI). It is believed that users will first focus on the ROI when they view an image, which means the quality of the ROI has a great impact on the overall quality assessment. When we remove the deformable convolution module from the local branch, we observe a decrease in the performance of all indicators in the first line of Table 3. For example, SROCC decreases by 6.35% from 0.9378 to 0.8818. It indicates the effectiveness of the deformable convolution module.

2) **Stereo cross attention module:** As binocular perception is a crucial factor to consider in SOIQA, the stereo cross attention module adopts the attention mechanism to simulate the interaction between the left and right views. To validate the effectiveness of this module, we remove the stereo cross attention module and directly

average the scores obtained from the left and right views. The final scores w.r.t. the three indicators are given in the second line of Table 3. Compared with the full model, the stereo cross attention module has achieved obvious performance gains in all evaluation metrics.

3) **Global and local branches:** As we employ a dual-stream framework, it is necessary to investigate the roles of each branch. To validate the effectiveness of the local branch, we remove the global branch and the subsequent feature fusion module. A similar process is performed for the validation of the global branch. According to Table 3, while the performance of the model with only the local branch shows a small decrease, i.e., PLCC decreases from 0.9342 to 0.9266, the performance of the model with only the global branch is significantly reduced. It shows that the local features of CMP images play a more important role in the assessment of SOI. Meanwhile, our integration of both ERP and CMP enables the complementary advantages of the two projection formats, collectively providing a complete visual perception of SOI.

## 5 CONCLUSION

In this study, we propose a dual-stream blind quality assessment model for stereoscopic omnidirectional images (SOI) that takes both the ERP images and CMP images as input, motivated by the observation that ERP provides global information and the less distorted CMP complements it by providing local structural guidance. A dual-stream backbone with Transformer and deformable CNN is designed to extract these global and local features simultaneously. Then an inter-intra feature fusion module is utilized to dynamically and complementarily integrate features from both projections, accounting for the inter-complementarity between ERP and CMP and the intra-relationships within CMP images. Lastly, with the features from left and right views of SOI, a stereo cross attention module that simulates the binocular fusion mechanism is proposed to predict the quality score. Extensive experiments are conducted on three publicly available databases, demonstrating that our proposed model outperforms the state-of-the-art competitors. In the future, we will continue on efficient global and local feature extraction and integration for SOI to further improve the performance of our model.

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
