# OpenReview forum: "Dual-stream Perception-driven Blind Quality Assessment for Stereoscopic Omnidirectional Images"
_acmmm.org/ACMMM/2024/Conference — MM2024 Poster_

### Official Review · Reviewer_B6bK · 2024-05-21

**Rating:** 3
**Confidence:** 3

**Summary:**

This paper presents a new stereoscopic omnidirectional image quality assessment (SOIQA) method, addressing the limitation of existing methods that rely solely on either Equirectangular Projection (ERP) or CubeMap Projection (CMP). The authors validated the effectiveness of proposed method through extensive experiments.

**Strengths:**

1.	The authors' proposition to leverage the complementary information from both ERP and CMP images is reasonable.
2.	The proposed model achieves the state-of-the-art on three public databases.

**Limitations:**

1.	The issue of the model's operational efficiency has a significant impact on the practical application of the model in real-world scenarios. While it is reasonable to utilize information from CMP, the proposed model also requires the generation of CMP images from ERP images during the inference process. To what extent does this affect the model's inference efficiency?
2.	The writing in the paper lacks precision, which is confusing. For example, the author claims that the proposed DSPDNet is "the first study" that leverages the complementary advantages of ERP and CMP, yet in line 285, the author mentions, "One recent interest in OIQA is how to utilize the complementarity of these two projection modes ERP and CMP." Despite differences in implementation, the literature [20] has already explored the use of ERP and CMP's complementarity for OIQA tasks.
3.	As shown in Table 3, the Global branch does not significantly contribute to the overall performance gain of the model, achieving only a very marginal improvement. Considering that this module introduces a Swin Transformer with a considerable number of parameters, such a performance gain is not satisfactory.
4.	The paper lacks references to the most recent literature. Besides the algorithms mentioned within the document, several new OIQA methods have been introduced. Unfortunately, these have not been cited, such as:

[1] Zhou W, Wang Z. Blind omnidirectional image quality assessment: integrating local statistics and global semantics[C]//2023 IEEE International Conference on Image Processing (ICIP). IEEE, 2023: 1405-1409.

[2] Zhou Y, Ding Y, Sun Y, et al. Perceptual Information Completion based Siamese Omnidirectional Image Quality Assessment Network[J]. IEEE Transactions on Instrumentation and Measurement, 2023.

**Suitability:**

2

---

### Official Review · Reviewer_EXGJ · 2024-05-25

**Rating:** 3
**Confidence:** 3

**Summary:**

This paper presents a dual-stream perception-driven BOIQA method, which uses both ERP and CMP features.

**Strengths:**

1. The organization of the paper is clear.
2. Many datasets have been used.

**Limitations:**

1. The used databases are not well referenced.
2. The authors don’t compare the method with other DNN-based models.
3. The proposed method lacks novelty, only the integration of ERP and cubic is not impressive.

**Suitability:**

2

---

### Official Review · Reviewer_f8fA · 2024-05-26

**Rating:** 5
**Confidence:** 2

**Summary:**

In this study, the authors propose a dual-stream blind quality assessment model for stereoscopic omnidirectional images (SOI), utilizing both equirectangular projection (ERP) and cubemap projection (CMP) images to capture global and local features, respectively. The model employs a dual-stream backbone with Transformer and deformable CNN, and an inter-intra feature fusion module to dynamically integrate features from both projections.

**Strengths:**

1.This paper is well-written, with no noticeable grammatical or spelling errors. The motivation is clearly articulated, highlighting the novelty of the SOIQA approach through dual collaboration across three levels: ERP and CMP projection images, CNN and transformer, and left-right views. The careful explanation by the authors lends rationality to the integration of these technologies, making it more than just a simple stacking of methods.

2.Introducing novel concepts like ERP and CMP to readers unfamiliar with the field is challenging. The authors successfully tackle this by thoroughly explaining the differences between the two projections in terms of their principles and characteristics, and justifying the rationale for utilizing both sets of information.

3.The experimental section is comprehensive, with the authors comparing SOIQA against a wide range of similar methods across multiple dimensions, demonstrating the robustness of their approach.

**Limitations:**

1.Despite the many strengths of this paper, some sections lack sufficient explanation, which might confuse readers. The abstract and the first two sections are well-developed but take up too much space. More emphasis could be placed on the model section, particularly on explaining the loss function used for SOIQA and clarifying its relation to DMOS values, which is currently unclear.

2.The authors justify the dual-stream approach using the human visual system, which includes left and right views. While distinguishing "depth information" through left and right eyes is common in VR games, it is not prevalent in the 3D image dataset field. This approach requires panoramic cameras to take pictures from two closely positioned but different locations. Even after reviewing the referenced datasets in this paper, I remain confused. Are there tools or algorithms that allow a single omnidirectional image to present left-right eye effects that the authors used but did not mention?

3.The use of ERP and CMP format images for feature extraction in the IQA field is innovative, although I have seen similar concepts in the super-resolution field (https://arxiv.org/abs/2112.06536). The key issue is that some technical details are not explained. For example, in Figure 3, the authors describe how features from the two projection types are fused and converted to the same dimension, but the challenges of this fusion are not addressed. It is known that there are tools to convert between ERP and CMP image formats, but the information they contain is still very different. Does the element-wise multiplication consider their gap? This is not an easy task. Although the paper discusses the necessity of doing so, unless I missed it, I did not see an explanation of how the challenges of this approach are overcome.

**Suitability:**

3

---

### Official Review · Reviewer_gTGi · 2024-05-29

**Rating:** 3
**Confidence:** 4

**Summary:**

This manuscript proposed a blind quality assessment method for stereoscopic omnidirectional images via integrating both ERP and CMP projections. Specifically, the inter-intra feature fusion module are designed to consider both the intercomplementarity between ERP and CMP and the intra-relationships within CMP images. Experimental results on the databases of LIVE 3D VR, SOLID, and NBU show that the proposed method achieves a superior performance.

**Strengths:**

- Exploiting both the global and local visual cues via integrating different projections such as ERP and CMP projections is reasonable. However, this concept is not new in 360 vision domain such as depth estimation and SOD tasks.
- Experimental results on the databases of LIVE 3D VR, SOLID, and NBU show that the proposed method achieves a superior performance, compared to the methods, including VP-BSOIQA[19], Chai-SOIQE[1] and SOIQE[4].

**Limitations:**

- **Incomplete literature review in related works.** In fact, the relevant work involved in this manuscript is not only a branch of the three IQA tasks of SIQA, OIQA, and SOIQA. To highlight the differences and connections between the innovation points proposed in this manuscript and existing related work, as well as technological contributions, a review should also be conducted on information fusion strategies in multi-projection domain for 360 vision tasks, such as depth estimation and salient object detection.
- **Experimental results are not convincing enough.** Although experimental results on the databases of LIVE 3D VR, SOLID, and NBU show that the proposed method achieves a superior performance, compared to the methods, including VP-BSOIQA[19], Chai-SOIQE[1] and SOIQE[4]. However, all of these methods were published before 2022 (including).
- **Lack of analysis on the motivation and contributions of proposed technique.**  Why can the proposed Dual-stream architecture learn quality-awared feature representation? More exploratory experiments are needed for detailed analysis and discussion. It is difficult to draw the conclusion that the proposed method is perception driven rather than data-driven based solely on the textual description in the current manuscript.

**Suitability:**

2

---

### Meta-Review · Area_Chair_i2R6 · 2024-07-07

**Recommendation:** Accept (Poster)
**Confidence:** 4

**Metareview:**

In the first round of reviews, the paper got 2 "borderline reject", 1 "borderline accept" and 1 "weak accept". After the authors submitted the rebuttal, one of the reviewers that leaned towards the rejection of this paper did not updated the review, while the other three reviewers lean towards the acceptance of this paper. Therefore, although the reviewers point out some limitations in novelty and some unclear issues with Figure 3, I would recommend this paper to be accepted as poster.